# Design and Experiment of the Buckwheat Hill-Drop Planter Hole Forming Device

Yu Chen, Yuming Cheng, Jun Chen *, Zhiqi Zheng, Chenwei Hu and Jiayu Cao

College of Mechanical and Electronic Engineering, Northwest A&F University, Xianyang 712100, China; jdxy73@nwafu.edu.cn (Y.C.); cym2018@nwafu.edu.cn (Y.C.); zhiqizheng@nwafu.edu.cn (Z.Z.); hcw@nwafu.edu.cn (C.H.); 18146842066@nwafu.edu.cn (J.C.)
* Correspondence: chenjun_jdxy@nwafu.edu.cn; Tel.: +86-29-8709-1867

**Abstract:** The hole forming device is an important element of the buckwheat hill-drop planter, and its design level directly affects the seeding quality of the hill-drop planter. A hole forming device with a duckbill structure is widely used in hill-drop planters for wheat, cotton, peanuts, etc. According to the requirements of buckwheat seeding operations, this study designs the components of the duckbill hole forming device. It is determined that the duckbill upper jaw length is 65 mm, the duckbills number is 10, the pressure plate on the spring side length is 90 mm, the duckbill opening size is 8.79 mm, and the duckbill effective opening time is 0.1 s. Through co-simulation analysis of discrete element software EDEM (DEM-Solutions, Edinburgh, United Kingdom) and multi-body dynamics software RecurDyn (FunctionBay, Inc., Seongnam-si, South Korea), it is measured that when the pressure plate on the spring side is directly below the rotation axis of the dibber wheel, the spring compression is 33.3 mm, the pressure on the pressure plate is 95–102.6 N, and the contact time of a single duckbill with the soil is 0.2 s at a speed of 40 r/min. Based on the results of the design and simulation analysis, the large end diameter, small end diameter, original length and wire diameter of the duckbill spring are 36 mm, 26 mm, 60 mm, and 1.8 mm, respectively. An experimental bench for the seeding wheel of a buckwheat hill-drop planter was built, and three wire diameter duckbill springs of 1.6 mm, 1.8 mm and 2.0 mm were tested to verify the simulation and calculation results. The experimental results show that the optimal wire diameter of the duckbill spring is 1.8 mm. Finally, a single factor experiment of the dibber wheel rotation speed was carried out. The experimental results show that when the rotation speed of the dibber wheel is 40–65 r/min, the seeding qualification rate, seeding void hole rate and seeding damage rate of the buckwheat hill-drop planter are ≥85.3%, 0, and <0.3%, respectively. This study provides a basis and reference for the hole forming device design of a buckwheat hill-drop planter.

**Keywords:** buckwheat; dibber wheel; hole forming device; EDEM; RecurDyn; co-simulation; experiment



## 1. Introduction

Buckwheat is a highly valuable nutritious crop [1,2], and it is also an important part of coarse cereals in many countries [3,4]. Due to the irregular shape of buckwheat seeds and the difficulty of mechanized seeding, the yield and production efficiency of buckwheat are low [5], which severely restricts the development of the buckwheat industry [6]. At present, there are three main seeding methods for buckwheat, including drill seeding, broadcast seeding and hole seeding [7,8]. Compared with drill seeding and broadcast seeding, hole seeding planting has the advantages of strong seedling breaking ability, fast seedling emergence, and high average yield [9], which meets the needs of precision agriculture operations. Therefore, it is of great significance for the buckwheat industry development to design a precise hill-drop planter with low manufacturing cost, fast sowing speed, easy popularization, and good universality.

The dibber wheel is the most important structural unit of a hill-drop planter, which is mainly composed of seeding apparatus and an acupoint hole forming device. The design level of the dibber wheel directly affects the seeding quality of the hill-drop planter. Therefore, related scholars and manufacturing companies have conducted a lot of research on the dibber wheel. After more than 70 years of development, the seed metering device-related technology tends to mature [10]. At present, seeding apparatus mainly includes mechanical seeding apparatus and pneumatic seeding apparatus, both of which have high working efficiency and a good precision seeding effect, and are widely used in the seeding process of rice and corn, etc. [11,12]. Parish designed a belt seeding apparatus to seed soybeans [13]. Wright developed a seed metering device to achieve precise control of the diamond-shaped cross arrangement of seeds and the hole spacing on the seed bed [14]. Sahoo optimized an inclined disc precision seed metering device [15]. Lei designed a pneumatic seeding apparatus. Through the co-simulation of Discrete Element Method DEM and Computational Fluid Dynamics CFD, a simulation experiment was conducted on the seeding process of pneumatic seeding apparatus, which provided a theoretical basis for the field test [16]. Ye designed an air-chamber rotary buckwheat seeding apparatus. A bench experiment was carried out with the qualified index, replay index, and missed seeding index as evaluation indexes, and the gas chamber vacuum degree, the seeding hole aperture, and the seeding disk rotation speed as test factors. The experimental results show that the designed buckwheat seeding apparatus is good, which meets the technical requirements of buckwheat precision seeding [17].

The hole forming device is an important device to ensure seeding performance parameters such as the seeding depth, hole distance, and seed numbers per hole. Currently, the hole forming device is mainly divided into a synchronous hole forming device and an asynchronous hole forming device. The synchronous hole forming device has advantages over the asynchronous hole forming device in precise hole spacing, so it is more widely used. The synchronous hole forming device is divided into a roller type hole forming device and a straight plug type hole forming device. The straight plug type hole forming device must be upright when it is put into the soil, which is generally used for planting seedlings. The roller type hole forming device includes duckbill type, shovel type and piston type. Hunt developed a precision hill-drop planter for vegetables, which adopted a piston type hole forming device. The seeding depth and hole distance could be flexibly adjusted by changing the mechanical structure of the piston and the number of pistons distributed on the disc [18]. FORIGO, a French company, has produced a film-covered hill-drop planter, which uses a duckbill device to punch holes in the film to complete hole seeding. SAMCO, an Irish company, has produced a film-covered hill-drop planter that also uses a duckbill device for drilling and hole seeding. At present, duckbill hole forming devices have been widely applied in the hill-drop planters of wheat and peanut, etc. [19,20]. Although the duckbill hole forming device has a simple structure and high reliability, due to the unique shape of buckwheat seeds and the agronomic requirements of buckwheat planting, it is necessary to design a hole forming device suitable for a buckwheat hill-drop planter to improve the precision and quality of buckwheat seeding.

In summary, this study is based on the requirements of buckwheat hill-drop seeding, and a duckbill hole forming device which has an important influence on the sowing accuracy and operation efficiency of a buckwheat hill-drop seeding machine was designed. Through co-simulation analysis of discrete element software EDEM (DEM-Solutions, Edinburgh, United Kingdom) [21–23] and multibody dynamics software RecurDyn (FunctionBay, Inc., Seongnam-si, South Korea) [24,25], the relevant mechanical data of duckbill's parts during the hill-drop planter operation were obtained. The duckbill spring is selected through theoretical analysis and numerical calculation, and the relevant parameters of the duckbill spring are obtained. A single factor experiment of the hole seeding wheel rotation speed was carried out to study the speed range that met the seeding industry standard of DG/T007-2019. This study provides basis and reference for hole forming device design of buckwheat hill-drop planter.

## 2. Materials and Methods

### 2.1. Design Requirements

According to agronomic requirements and related buckwheat planting techniques, buckwheat dibber seeding depth is generally 30–50 mm, row spacing is 330 mm, hole spacing is 170–200 mm, with 5–7 buckwheat seeds per hole. With reference to the local standards of China's Shanxi, Shaanxi, Gansu and other provinces, the main technical parameters of the buckwheat hill-drop planter are determined as shown in Table 1.

**Table 1.** Main technical parameters of buckwheat hill-drop planter.

| Overall Dimensions (mm) | Seeding Row Number | Seeding Type | Operation Row Spacing (mm) | Hole Spacing (mm) | Seeding Depth (mm) | Seeds Number per Hole | Driving Speed (km/h) |
|---|---|---|---|---|---|---|---|
| L × W × H: 2000 × 2500 × 1500 | 4 | hole seeding | 380 ± 20 | 170 ± 10 | 20–40 | 3–5 | 4–6 |

### 2.2. Design and Analysis of Key Components of the Hole Forming Device

The duckbill hole forming device is widely used in the hill-drop planter of wheat, cotton, peanut, sugar beet, etc. [26,27]. The duckbill hole forming device structure is shown in Figure 1a. The main structure includes a pressure plate (1), a return spring (2), a fixed plate (3), a fixed pin (4), and a duckbill upper jaw (5). The fixed plate (3) is connected to the outer circle of the seed metering device (6). The return spring (2) is fixed between the pressure plate (1) and the fixed plate (3). The pressure plate (1) is connected to the duckbill upper jaw (5) through the fixed pin (4). When the dibber wheel rotates, the duckbill upper jaw (5) first touches the ground and is inserted into the soil, then the pressing plate (1) touches the ground and rotates around the fixed pin (4) to make the duckbill open to a certain size for seeding. At the same time, the return spring (2) is compressed. Due to the spring force, the duckbill (5) is closed when the pressure plate (1) leaves the ground. The main design parameters of the duckbill include the fixed plate bending radius, duckbill number, the hole depth formed by the duckbill, the duckbill upper jaw inclination angle, the width and length of the duckbill, etc.

In Figure 1b, R is the seed metering device radius (mm), L is the distance from the center of the dibber wheel to the top of the duckbill upper jaw (mm), and θ is the angle between two adjacent duckbill upper jaws (°).

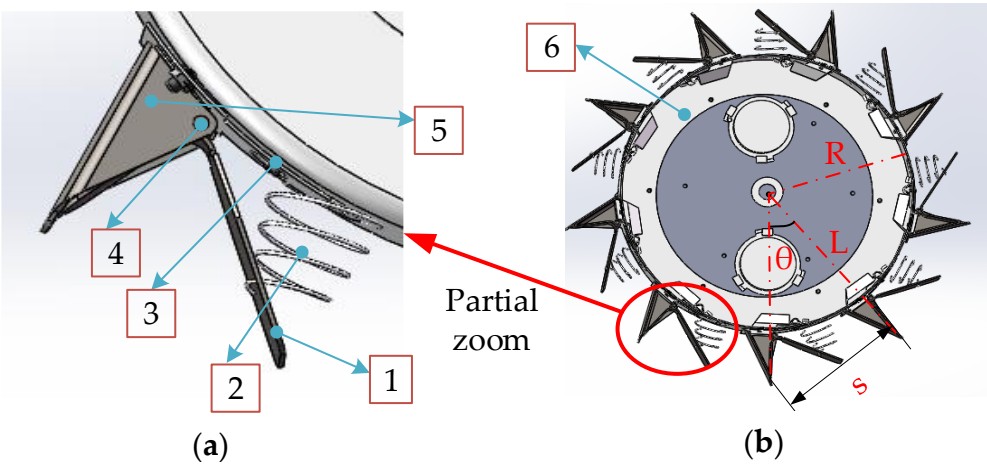

**(a)**　　　　　　　　　　　　　　　　　　　　**(b)**

**Figure 1.** Structure schematic diagram of dibber wheel and duckbill. (**a**) Duckbill; (**b**) dibber wheel.

#### 2.2.1. Determination of the Duckbill Upper Jaw Length

According to Table 1, it is determined that the seed metering device radius is 210 mm, the seeding depth is 30 mm, and the seeding hole distance is 170 ± 10 mm. The seeding

hole distance refers to the distance between two duckbill upper jaws in Figure 1b. The seeding hole distance is set to be 170 mm. The duckbills number is 10, evenly distributed on the outside of the seed metering device. The duckbill length $L_1$ can be determined by Formulas (1)—(3). In Formulas (1)—(3), Z is the duckbills number, and s is the seeding hole distance (mm).

$$\theta = \frac{360}{Z} \tag{1}$$

$$L = \frac{s}{2\sin\frac{\theta}{2}} \tag{2}$$

$$L_1 = L - R \tag{3}$$

Through calculation, the duckbill length $L_1$ = 65 mm.

### 2.2.2. Pressure Plate Design

When the dibber wheel is not working, the duckbill upper jaw and the end of the pressure plate are designed to be on the same circle, as shown in Figure 2a, which is to ensure that when the dibber wheel rotates to the same position, the depth of the duckbill upper jaw into the soil and the amount of compression at the end of the pressure plate on the spring side are equal. According to buckwheat planting requirements, the seeding depth is set to 30 mm. When the dibber wheel is seeding, the duckbill upper jaw and the pressing plate alternately form a polygonal movement of the main fulcrum. In order to stabilize the seeding depth at 30 mm, 10 duckbill upper jaws must be on the outer circle as shown in Figure 2b. At the same time, in order to ensure the dibber wheel rotates smoothly, the ends of the 10 pressure plates should be compressed by 30 mm when they are rotated to the bottom end. After calculation, the length of the pressure plate on the spring side is 90 mm.

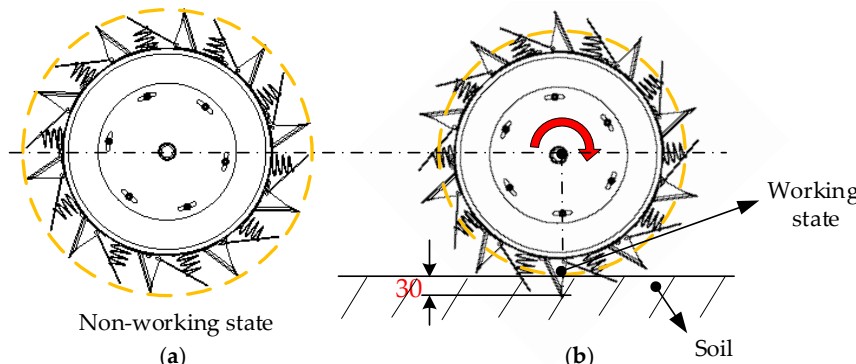

**Figure 2.** Schematic diagram of the duckbill non-working state and working state outer circle. (**a**) Schematic diagram of the duckbill non-working state outer circle; (**b**) Schematic diagram of the duckbill working state outer circle.

Figure 3 depicts the four positions of a duckbill in contact with the soil during the dibber wheel rotation. Figure 3a is the moment the duckbill upper jaw touches the soil. Figure 3b is the moment the pressure plate on the spring side touches the soil. Figure 3c is the deepest moment when the duckbill upper jaw enters the soil. Figure 3d is the maximum moment when the spring is compressed.

According to the design parameters in Table 1, the seeding depth is 20–40 mm. In order for the buckwheat hill-drop planter to discharge seeds smoothly and meet the requirements of seeding depth, the following conditions must be met.

(1) When the dibber wheel is in the position shown in Figure 3b, the pressure plate has just touched the soil, and the vertical depth of the duckbill upper jaw has reached 20 mm, which can ensure that the depth of the seed will not be less than 20 mm during seeding.

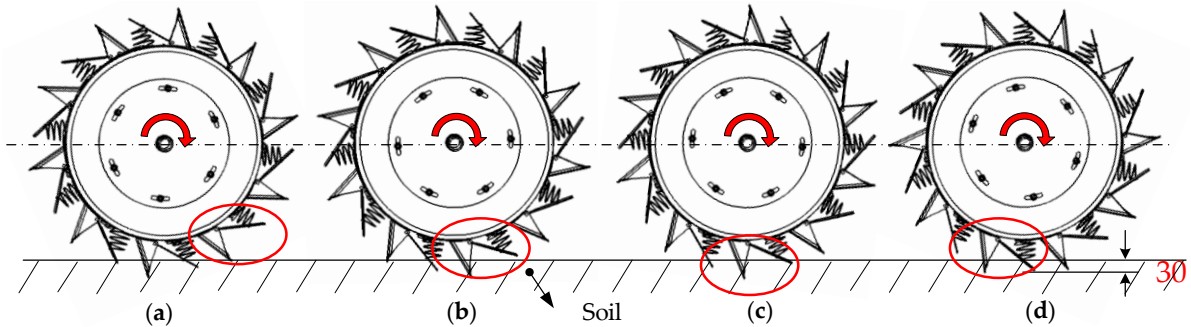

**Figure 3.** (**a**–**d**) Four places where the duckbill touches the soil.

(2) When the duckbill upper jaw reaches the deepest seeding depth shown in Figure 3c, the opening size of the duckbill must meet the requirements for smooth seeding; that is, the opening size is 1.2–1.5 times the maximum length of the of buckwheat three-axis dimension. Assuming that the maximum length of the of buckwheat three-axis dimension is $a_{max}$, the value range of the duckbill opening K is $1.2\,a_{max} \leq K \leq 1.5\,a_{max}$. Through existing research, the value range of K is determined to be 7.74–9.675 mm.

Figure 4 shows the process of the pressure plate rotating from A to B when the dibber wheel rotates from the position shown in Figure 3b,c. Combining Equations (4)–(6), it can be obtained that the pressure plate has rotated the angle $\theta_1$ by 7.2°, and the opening size K of the pressure plate on the seeding side is 8.79 mm, which meets seeding conditions of buckwheat hill-drop planter.

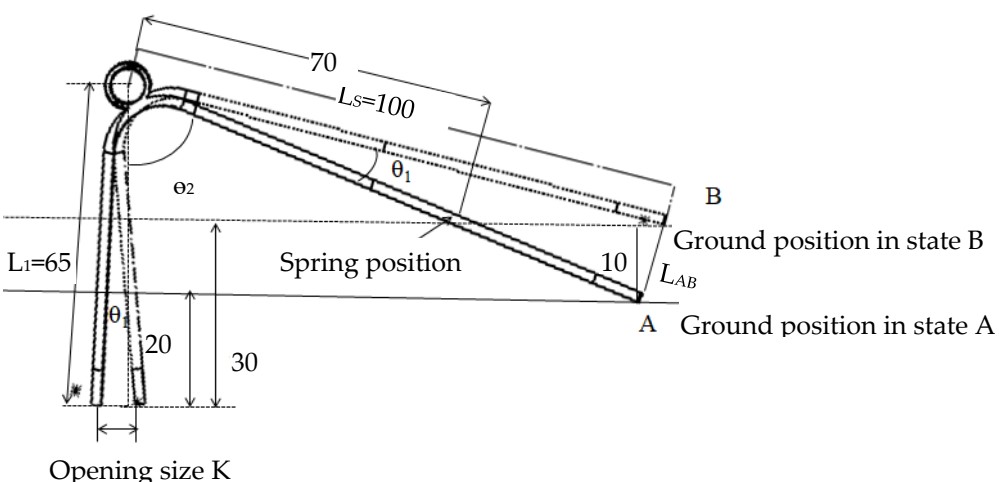

**Figure 4.** Schematic diagram of pressure plate rotation.

$$\theta_1 = 2\arcsin\frac{K}{2L_1} \tag{4}$$

$$\theta_1 = \arccos\left(\frac{65-30}{90}\right) - \arccos\left(\frac{65-20}{90}\right) \tag{5}$$

$$L_{AB} = 2L_S \sin\frac{\theta_1}{2} \tag{6}$$

In Formulas (4)–(6), $L_1$ is the pressure plate on the seeding side (mm); $L_S$ is the length of the pressure plate on the spring side (mm); and $L_{AB}$ is the rotation distance at the end of the pressure plate on the spring side (mm).

Figure 5 shows the seed position when the duckbill is closed. In the case that the opening size of the duckbill meets the requirements, it should also be considered that the time t for the seeds to fall from the duckbill to the seeding hole is less than the effective time T for the duckbill to open, that is, t < T. The effective time of duckbill opening refers

to the time when the duckbill opening size K $\geq$ 7.74 mm under the premise that the depth of the duckbill's penetration into the soil is in the range of 20–40 mm.

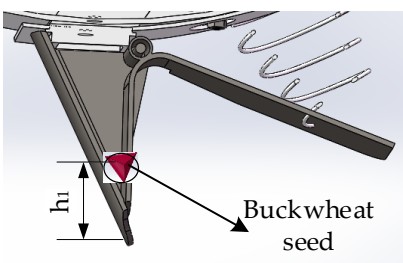

**Figure 5.** Buckwheat seed location diagram.

When the rotational speed of the dibber wheel is 40 r/min, it can be obtained that t = 0.041 s through Equations (7)–(10). According to Equation (11), T = 0.10 s can be obtained, which satisfies the condition of smooth seeding. According to Formulas (7)–(10), when the rotational speed of the dibber wheel is 40–80 r/min, all of them satisfy t < T.

$$h_1 = \frac{1}{2}at^2 \tag{7}$$

$$G_1 + F = ma_g \tag{8}$$

$$F = m\frac{v^2}{R} \tag{9}$$

$$v = wR = R\frac{2\pi n}{60} \tag{10}$$

$$T = \varphi_1\frac{60}{360n} \tag{11}$$

In Formulas (7)–(11), $a_g$ is seed acceleration (m/s$^2$); F is the centrifugal force of the seed (N); n is the rotational speed of the dibber wheel (r/min); and $\varphi_1$ is the rotation angle required by seeding. According to Formulas (7)–(11), the value of $\varphi_1$ is 24°.

### 2.3. Co-Simulation Analysis of EDEM and RecurDyn
#### 2.3.1. The Influence of Spring on Cavitation Hole Forming

The factors affecting the seeding qualification of the buckwheat hill-drop planter are not only the qualified rate of hole number, but also the qualified rate of seeding depth, qualified rate of hole spacing, and qualified rate of row spacing. The design needs to pay attention to the influence of hole spacing, hole seeding depth (duckbill upper jaw depth into soil) and duckbill opening size on the smooth discharge of seeds. The spring has an important effect on the seeding depth and the size of the duckbill opening. When the spring stiffness is too large, the pressure plate opening is too small and the seeds cannot be discharged smoothly. When the spring stiffness is too small, the seeding depth will not be reached, and the duckbill will open in advance to expel the seeds. In order to obtain the spring parameters, a co-simulation model of EDEM and RecurDyn was established to simulate the force of the spring on the pressure plate and the force of the duckbill device in the field work environment. In turn, parameters such as the stiffness coefficient and the wire diameter of the spring are obtained, which provide reliable data for the bench test of the duckbill device. Since EDEM software cannot perform complex multi-body dynamics analysis, and RecurDyn cannot generate particles, it is necessary to establish a co-simulation model of EDEM and RecurDyn.

### 2.3.2. Establishment of EDEM Simulation Model

Solid Works software was used to draw the seed metering device model of the buckwheat dibber wheel, as shown in Figure 6a. The model was imported into EDEM 2.7 software [28–30], and the simulation model was generated, as shown in Figure 6b.

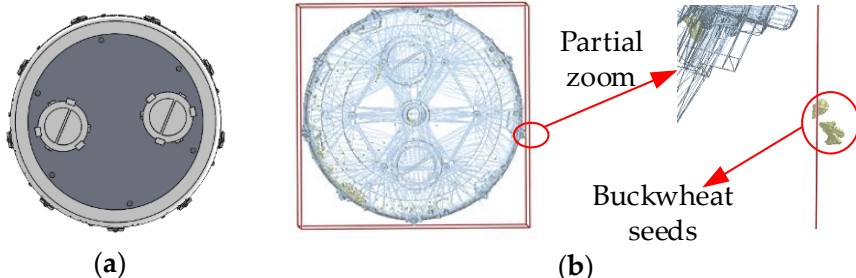

(**a**)                    (**b**)

**Figure 6.** EDEM simulation model of seed metering device. (**a**) 3D model; (**b**) grid structure simulation model.

Combining the three-axis size of buckwheat seeds and the special shape of buckwheat grains, the ball polymerization method was used to simplify the processing of buckwheat seed particles [31,32]. The length, width and thickness of buckwheat seeds were used as parameters of the normal distribution of particle factory size. In order to simulate the real shape of buckwheat, 44 spheres were selected for simulation, and modeling was carried out according to the size of buckwheat seeds actually measured. The particle model is shown in Figure 7.

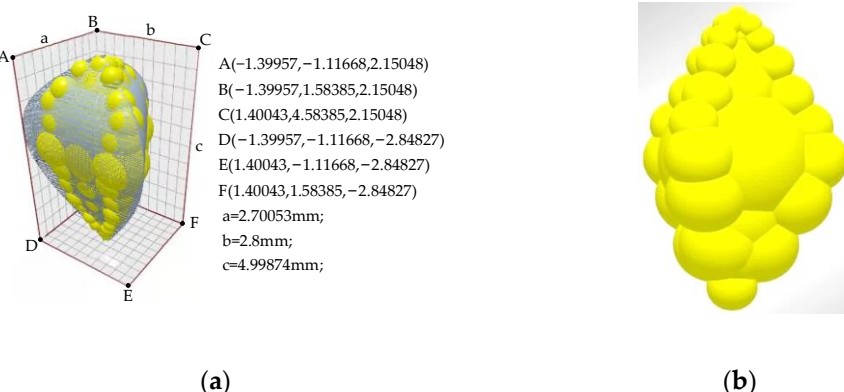

(**a**)                    (**b**)

**Figure 7.** Particle model of buckwheat. (**a**) Schematic diagram of particle filling; (**b**) particle model.

### 2.3.3. RecurDyn Simulation Analysis Principle

RecurDyn multi-body dynamics analysis includes two stages of modeling and solving. The multi-body dynamic relationship model is formed by imposing kinematic constraints, driving constraints, force loads, initial condition calculations, etc., and the motion simulation is completed by determining relative kinematic coordinates, constructing relative motion, speed, and force.

(1) Relative kinematics coordinate system. The rigid body moves in the three-dimensional space, and the corresponding rectangular coordinate system is established with O as the coordinate origin, and the coordinate expression of point A is shown in Formula (12). The relative mathematical expression of any point P on the rigid body relative to the coordinate origin O can be expressed as the coordinates shown in Figure 8.

$$A = \begin{bmatrix} a_{11} & a_{12} & a_{13} \\ a_{21} & a_{22} & a_{23} \\ a_{31} & a_{32} & a_{33} \end{bmatrix} = \begin{bmatrix} f & g & h \end{bmatrix} \tag{12}$$

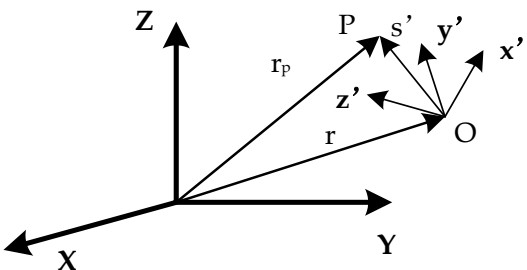

**Figure 8.** Coordinate diagram.

In Equation (12) and Figure 8 f–h there are unit vectors along the **x′**, **y′**, and **z′** axes, respectively. **x′-y′-z′** is the rigid body conjoined coordinate system. **X-Y-Z** is the inertial reference coordinate system.

The velocity and virtual displacement of point O in the **X-Y-Z** coordinate system were defined as Equations (13) and (14), respectively.

$$\left[ \begin{array}{c} \dot{r} \\ w \end{array} \right] \tag{13}$$

$$\left[ \begin{array}{c} \delta_r \\ \delta_p \end{array} \right] \tag{14}$$

The velocity and virtual displacement of point O in **x′-y′-z′** are defined as Equations (15) and (16).

$$\left[ \begin{array}{c} \dot{r}' \\ w' \end{array} \right] = \left[ \begin{array}{c} A^T_{\dot{r}} \\ A^T_w \end{array} \right] \tag{15}$$

$$\left[ \begin{array}{c} \delta'_r \\ \delta'_\pi \end{array} \right] = \left[ \begin{array}{c} A^T \delta_r \\ A^T \delta_\pi \end{array} \right] \tag{16}$$

In order to constructed relative motion, Assuming component $(i-1)$ is the previous component of component $(i)$, the coordinates of point Oi are expressed as Equation (17).

$$r_i = r_{(i-1)} + s_{(i-1)i} + d_{(i-1)i} - s_{i(i-1)} \tag{17}$$

To define $A_{(i-1)i} = A^T_{(i-1)} A_i$, the Formula (17) can be used to calculate the angular velocity of member i as Formula (18), and its reference coordinate system is located inside itself.

$$w'_i = A^T_{(i-1)i} w'_{(i-1)} + A^T_{(i-1)i} H'_{(i-1)i} \dot{q}_{(i-1)i} \tag{18}$$

In Formula (18), H is determined by the rotation axis.

(2) Speed recursive equation

Since the calculations of the speed recursive method and the speed transformation method are equivalent, for any $x \in Rnr$, Equations (19) and (20) are established:

$$X = B\dot{x} \tag{19}$$

$$X_i = B_{(i-1)i1} X_{(i-1)} + B_{(i-1)i2} x_{(i-1)} \tag{20}$$

where $x \in Rnr$ is obtained by multiplying B and x. Using Equations (19) and (20) iteratively, when $x \in Rnr$ is transformed into $Bx \in Rnr$, the calculation is more efficient.

In the force recursive algorithm, the recursive algorithm equation of Q* is shown in Equation (21):

$$\delta W = \sum_{i=0}^{n-1} \delta q^T_{i(i+1)} \left\{ B^T_{i(i+1)2} (Q_{i+1} + S_{i+1}) \right\} \tag{21}$$

where,

$$S_0 = 0$$
$$S_{i+1} = B_{(i+1)(i+2)1}^T (Q_{i+2} + S_{i+2}) \qquad (22)$$

Incorporating Equation (22) into Equation (21), the recursive algorithm equation of Q*
can be obtained as shown in Equation (23):

$$Q_{i(i+1)}^* = B_{(i+1)(i+2)2}^T (Q_{i+1} + S_{i+1}), i = n - 1, \ldots, 0 \qquad (23)$$

### 2.3.4. Co-Simulation of EDEM and RecurDyn

The 3D model of the dibber wheel was imported into RecurDyn in ".step" format, and
the corresponding constraints were added to each part, including the effect of applied force
(such as gravity, spring force and other interaction forces), kinematic constraints (joint),
and driving constraints. Initial conditions and corresponding contact relations were set,
and the dynamics model was generated as shown in Figure 9.

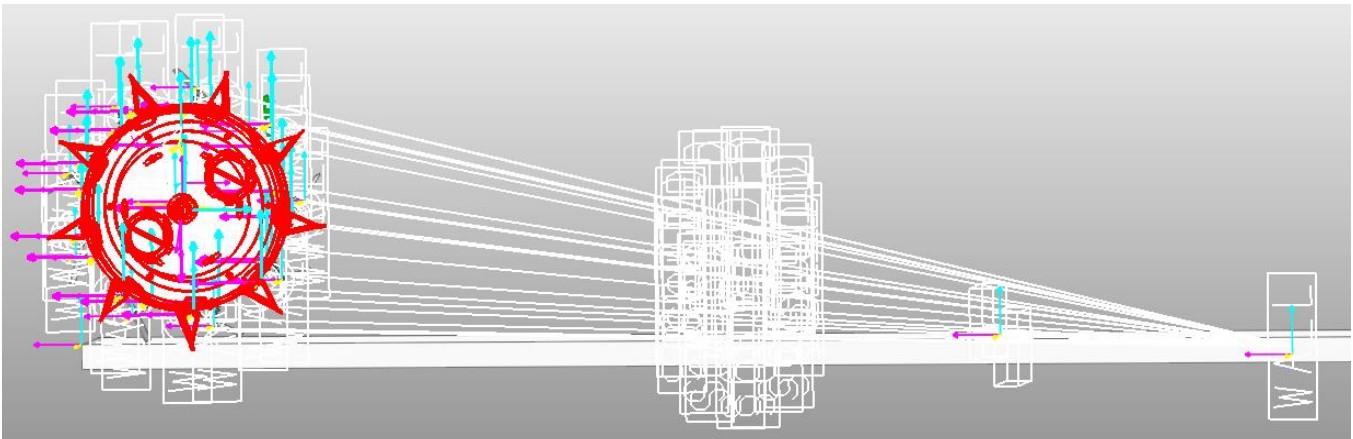

**Figure 9.** RecurDyn dynamics simulation model.

Set simulation parameters such as spring parameters, soil parameters and seed parameters, as shown in Table 2. After the parameter setting was completed, the simulation
analysis was started. If there was no error, the moving parts of the model were imported
into EDEM, and the particle factory was generated in EDEM.

**Table 2.** Simulation parameter table.

| Parameter Types | Parameters | Values |
|---|---|---|
| Spring parameters | Stiffness coefficient (N/mm) | 1–3 |
| | Damping coefficient | 0.035 |
| | Free length (mm) | 70 |
| | Spring diameter (mm) | 50 |
| | Coil number | 6 |
| | Wire diameter (mm) | 1.8 |
| Contact parameters | Stiffness coefficient (N/mm) | 100 |
| | Damping coefficient | 0.03 |
| | Dynamic friction coefficient | 0.01 |
| | Stiffness index | 2 |
| | Rebound damping coefficient | 0.25 |
| Soil parameters | Relative density (kg/m$^3$) | 1850 |
| | Shear modulus (Pa) | $10^6$ |
| | Poisson's ratio | 0.38 |

**Table 2.** *Cont.*

| Parameter Types | Parameters | Values |
|---|---|---|
| Duckbill parameters | Density (kg/m$^3$) | 7850 |
| | Shear modulus (Pa) | $8.23 \times 10^{10}$ |
| | Poisson's ratio | 0.3 |
| Interaction parameters | Soil–soil collision recovery coefficient | 0.66 |
| | Soil–duckbill collision recovery coefficient | 0.51 |
| | Soil–soil static friction coefficient | 0.83 |
| | Soil–duckbill static friction coefficient | 0.5 |
| | Soil–soil dynamic friction coefficient | 0.25 |
| | Soil–duckbill dynamic friction coefficient | 0.05 |
| Other parameters | Total number of particles | 100,000 |
| | Acceleration of gravity (m/s$^2$) | 9.81 |
| | Time step (s) | $3.14097 \times 10^{-5}$ |

### 2.4. Bench Test

As shown in Figure 10, in order to verify the accuracy of the duckbill spring parameters calculated by the above simulation and theoretical calculations, three spring diameters of 1.6 mm, 1.8 mm, and 2.0 mm were selected to test Yuqiao No. 3 sweet buckwheat, and the qualified rate of seeding and hole spacing of the buckwheat dibber wheel was carried out.

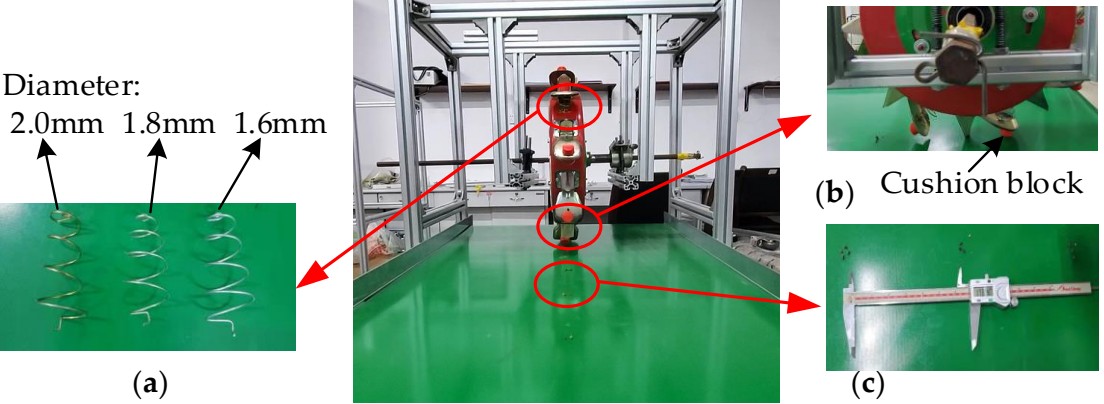

**Figure 10.** Bench test. (**a**) Three springs with different diameters; (**b**) Cushion block; (**c**) Measurement of hole spacing.

Three repetitive tests were carried out on springs with three wire diameters, and the rotation speed of the dibber wheel was 40 r/min. During the bench test, the duckbill upper jaw could not be inserted into the soil by 30 mm. In order to simulate the opening size of the duckbill and the state of the duckbill when it reached the seeding depth, a cushion block was installed at the position of Figure 10b, and the cushion block height was 30 mm.

### 2.5. Experiment on Rotation Speed and Seeding Qualified Rate of the Dibber Wheel

In order to further study the change law of the designed dibber wheel seeding performance with its rotation speed, the maximum rotation speed of the dibber wheel was explored when the hole number qualified rate of the hill-drop planter is ≥85%, the void hole rate is ≤2%, and the damage rate is ≤1.5%. Taking Figure 10 as the test platform, the seeding performance of the dibber wheel was studied when the rotation speed of the dibber wheel was 40 r/min, 45 r/min, 50 r/min, 55 r/min, 60 r/min, and 65 r/min. The experiment indexes were seeds number per hole qualified rate $y_1$, void hole rate K, and damage rate P, and the calculation formulas were shown in Formulas (24)–(26).

$$y_1 = \frac{m_1}{m} \times 100\% \tag{24}$$

$$K = \frac{m_k}{M} \times 100\% \tag{25}$$

$$P = \frac{B_1 - B_0}{Z_1} \times 100\% \tag{26}$$

In Formulas (24)–(26), $m_k$ is the number of void holes; M is total number of measuring holes; $B_1$ is the seed damage rate in the seed box; $B_0$ is the original seed damage rate; and $Z_1$ is the total number of seeds seeding.

## 3. Results and Discussion

### 3.1. Co-Simulation Results of EDEM and RecurDyn

When the rotation speed of the dibber wheel is 40 r/min, the coupling results of EDEM and RecurDyn are shown in Figure 11, in which Figure 11a is the schematic diagram of the overall simulation. Figure 11b–d are partial enlarged views of the duckbill in the simulation process, which are the position where the duckbill upper jaw starts to touch the soil, the position where the pressure plate just touches the soil, and the position where the duckbill upper jaw leaves the soil.

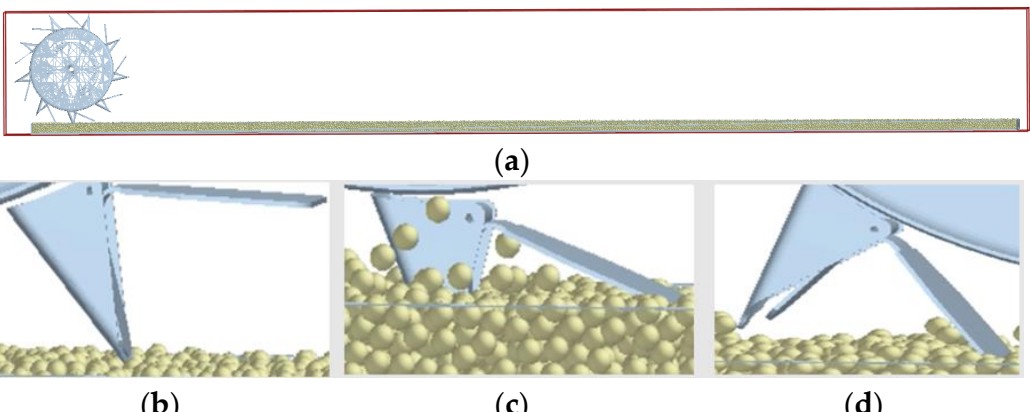

**(a)**

**(b)**          **(c)**          **(d)**

**Figure 11.** Co-simulation results of EDEM and RecurDyn. (**a**) Schematic diagram of the overall simulation; (**b**) The position where the duckbill upper jaw starts to touch the soil; (**c**) The position where the pressure plate just touches the soil; (**d**) The position where the duckbill upper jaw leaves the soil.

The simulation can analyze the force of the pressure plate on the spring side when each duckbill upper jaw reaches directly below the rotating shaft during the rotation of the dibber wheel, and the maximum value of the centrifugal force on the pressure plate during the movement of the non-contact soil. Figure 12 shows the simulation results of the pressure on the plate at the lowest point. The results show that when the dibber wheel rotation speed is 40 r/min and the seeding depth of the duckbill upper jaw reaches 30 mm, the maximum pressure F from the dibber wheel on the pressure plate is 95–102.6 N, which is affected by gravity and centrifugal force in the non-contact soil working area. The maximum resultant force $F_1$ is 1.311 N, and the total time for the duckbill upper jaw to contact the soil is 0.20 s.

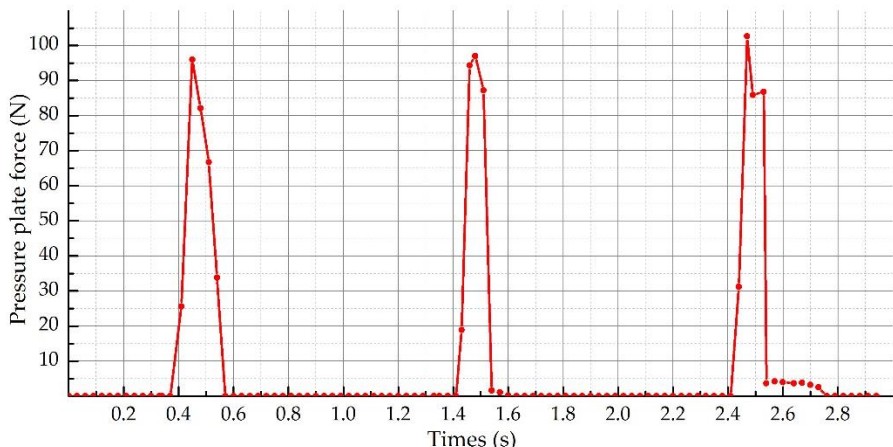

**Figure 12.** Simulation results of pressure plate force with RecurDyn.

Through analysis, when the spring is in the position of Figure 3d, the pressure plate is located directly below the rotation axis, the force and deformation $\Delta L_2$ of the spring are the largest, and its value is 23.3 mm. According to the relative dimensions of the spring position shown in Figure 13, the original length L of the design spring is 60 mm. When the duckbill is not in the soil, the compression amount $\Delta L_1$ of the spring is 10 mm. Therefore, the maximum spring compression $\Delta L$ during the rotation of the dibber wheel is 33.3 mm. It is known that the F value is 95–102.6 N through simulation. Substituting F and $\Delta L$ into Equation (27), the spring stiffness coefficient c can be calculated to be 2.853–3.081 N/mm.

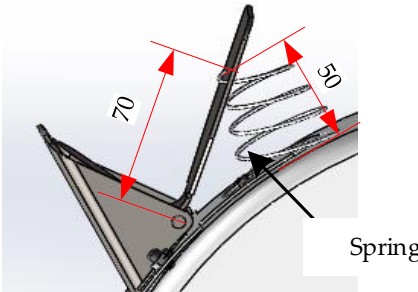

**Figure 13.** Spring position diagram.

$$F \;=\; c_K \Delta L = \frac{Gd^4 \left(R_2 - R_1\right)}{16N_1 \left(R_2^4 - R_1^4\right)} \Delta L \tag{27}$$

In Formulas (27), $c_k$ is the stiffness coefficient of the spring; F is the load on the spring (N); $\Delta L$ is the spring shape variable (mm); and G is the shear modulus of spring material (MPa). When the spring material is carbon steel, G = 79,000 MPa. When the spring material is stainless steel, G = 71,000 MPa. d is the spring wire diameter (mm). $R_1$ and $R_2$ are half of the small end middle diameter and half of the big end middle diameter of the spring, respectively (mm). $N_1$ is the number of spring coils.

The spring stiffness coefficient is directly related to the wire diameter, the material shear modulus, the turns number that the spring participates in the work, and the major and minor diameters of the spring. The spring material is general carbon steel, the large end middle diameter of the spring is 36 mm, the small end middle diameter of the spring is 26 mm, and the coils number is 3.5. Through simulation and theoretical calculations, it can be determined that the optimal value of the spring wire diameter d is 1.795–1.8523, so the wire diameter of 1.8 mm is taken as the optimal solution obtained by simulation and theory. Figure 14a–c show the original state of the spring, the state after the pressure plate is installed, and the state at the time of maximum compression deformation, respectively.

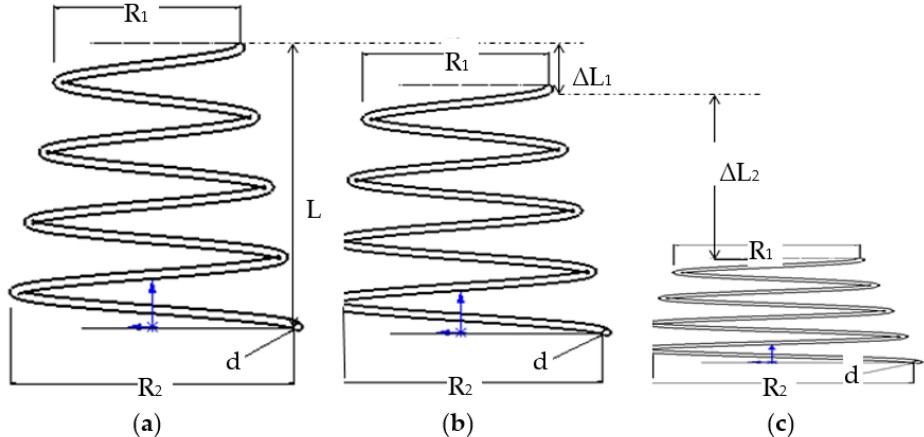

**Figure 14.** Simulation schematic diagram of the spring force and deformation with RecurDyn. (**a**) Spring original state; (**b**) The state after the pressure plate is installed; (**c**) The state at the time of maximum compression deformation.

### 3.2. Results of the Bench Test

A vernier caliper was used to measure the hole distance of the dibber wheel, as shown in Figure 10c. The results of the qualification rate are shown in Table 3. If the amount of buckwheat per hole is less than or equal to two grains, it can be considered as missed seeding. If the amount of buckwheat per hole is more than six grains, it can be considered as repeat seeding. If there are three to five grains of buckwheat per hole, it is considered qualified.

**Table 3.** Statistical table of bench test qualification rate.

| Parameters | | Number of Test Samples (100 holes) | | | Average Qualification Rate (%) |
|---|---|---|---|---|---|
| The Spring Wire Diameter (mm) | Test Number | Miss-Seeding Rate (%) | Repeat Seeding Rate (%) | Qualification Rate (%) | |
| 1.6 | 1 | 4 | 7 | 89 | |
| 1.6 | 2 | 4 | 6 | 90 | 89.33 |
| 1.6 | 3 | 5 | 8 | 89 | |
| 1.8 | 4 | 3 | 6 | 91 | |
| 1.8 | 5 | 4 | 5 | 91 | 91.67 |
| 1.8 | 6 | 3 | 4 | 93 | |
| 2.0 | 7 | 5 | 6 | 89 | |
| 2.0 | 8 | 6 | 8 | 86 | 88.00 |
| 2.0 | 9 | 4 | 7 | 89 | |

Table 4 shows the statistical results of buckwheat dibber wheel hole distance. The test number is the same as that in Table 3. Ten groups of data are measured in each test.

**Table 4.** Hole distance statistics in bench test.

| Test Number | Hole Distance/mm | | | | | | | | | |
|---|---|---|---|---|---|---|---|---|---|---|
| | 1 | 2 | 3 | 4 | 5 | 6 | 7 | 8 | 9 | 10 |
| 1 | 173.44 | 172.22 | 170.74 | 176.04 | 170.96 | 171.10 | 173.90 | 176.30 | 178.20 | 178.90 |
| 2 | 169.82 | 170.68 | 168.34 | 174.24 | 172.38 | 170.18 | 169.56 | 172.08 | 169.54 | 170.28 |
| 3 | 168.74 | 170.06 | 171.28 | 169.32 | 171.24 | 170.82 | 169.06 | 172.24 | 169.24 | 170.38 |
| 4 | 172.68 | 169.96 | 174.36 | 169.94 | 169.98 | 173.82 | 168.72 | 169.96 | 170.54 | 173.56 |
| 5 | 169.88 | 168.86 | 168.68 | 168.90 | 168.70 | 174.42 | 167.38 | 168.88 | 170.36 | 174.28 |
| 6 | 163.64 | 168.32 | 175.32 | 171.28 | 172.96 | 169.92 | 169.90 | 174.40 | 173.48 | 169.98 |
| 7 | 170.46 | 167.60 | 171.24 | 173.36 | 168.80 | 171.28 | 173.24 | 166.82 | 172.46 | 168.86 |
| 8 | 172.34 | 169.40 | 172.20 | 178.64 | 174.40 | 171.16 | 169.32 | 172.56 | 170.20 | 172.46 |
| 9 | 169.66 | 170.02 | 169.96 | 178.66 | 172.20 | 173.18 | 172.46 | 169.36 | 172.38 | 171.76 |

The experimental data in Table 4 were analyzed, and Equations (28)–(30) were used to process the data. The results are shown in Table 5.

$$\overline{X} = \frac{\sum\limits_{i=1}^{n} X_i}{n} \tag{28}$$

$$SD = \frac{\sqrt{\sum\limits_{i=1}^{n} (X_i - \overline{X})^2}}{n-1} \tag{29}$$

$$CV = \frac{SD}{\overline{X}} \tag{30}$$

**Table 5.** Data analysis of test index.

| The Spring Wire Diameter (mm) | Test Index | Rotational Speed of the Dibber Wheel (r/min) | Mean Value (mm) | Standard Deviation | Coefficient of Variation (%) | Qualification Rate (%) |
|---|---|---|---|---|---|---|
| 1.6 | | | 171.71 | 2.72 | 1.58 | 100% |
| 1.8 | Hole distance | 40 | 170.77 | 2.65 | 1.55 | 100% |
| 2.0 | | | 171.55 | 2.64 | 1.54 | 100% |

As shown in Table 5, when the rotation speed of the dibber wheel was 40 r/min and the spring wire diameter was 1.8 mm, the qualification rate of buckwheat number per hole in the bench test was the best, and the mean value of the three tests was 91.67%, which was consistent with the results of the simulation test and theoretical calculation. At this point, the mean hole distance was 170.77 mm, the standard deviation was 2.65, the coefficient of variation was 1.55%, and the qualification rate of hole distance was 100%. The structure and size of each part of the duckbill design is reasonable, which conforms to the parameter design of the hill-drop planter and the agronomic requirements of buckwheat seeding.

*3.3. Experimental Results of Rotation Speed and Seeding Qualification Rate in the Dibber Wheel*

The experimental results of rotation speed and seeding qualification rate in the dibber wheel is shown in Table 6.

**Table 6.** Statistic of bench test results.

| Test Number | Parameters | Average Value of Three Repeated Tests | | |
|---|---|---|---|---|
| | Rotational Speed of the Dibber Wheel (r/min) | Qualification Rate (%) | Void Hole Rate (%) | Damage Rate (%) |
| 1 | 40 | 92.33 | 0 | 0.12 |
| 2 | 45 | 90.67 | 0 | 0.09 |
| 3 | 50 | 89.00 | 0 | 0.07 |
| 4 | 55 | 87.67 | 0 | 0.06 |
| 5 | 60 | 86.33 | 0 | 0.17 |
| 6 | 65 | 85.33 | 0 | 0.23 |
| 7 | 70 | 79.67 | 0 | 0.36 |

Figure 15 shows that when the rotation speed of the dibber wheel changes between 40–65 r/min, the seeding qualification rate gradually decreases with the increase of the rotation speed. The seeding qualification rate is greater than 85% when the rotation speed of the dibber wheel changes between 40–65 r/min. The seeding qualification rate meets the requirements of planter operations. When the rotation speed of the dibber wheel changes between 40–70 r/min, the seeding damage rate is about 0.1%, and all of them are less than

1.5%. Table 6 shows that the void hole rate in the whole test is 0, indicating that there will be less seeding during the test, but no void hole will be caused.

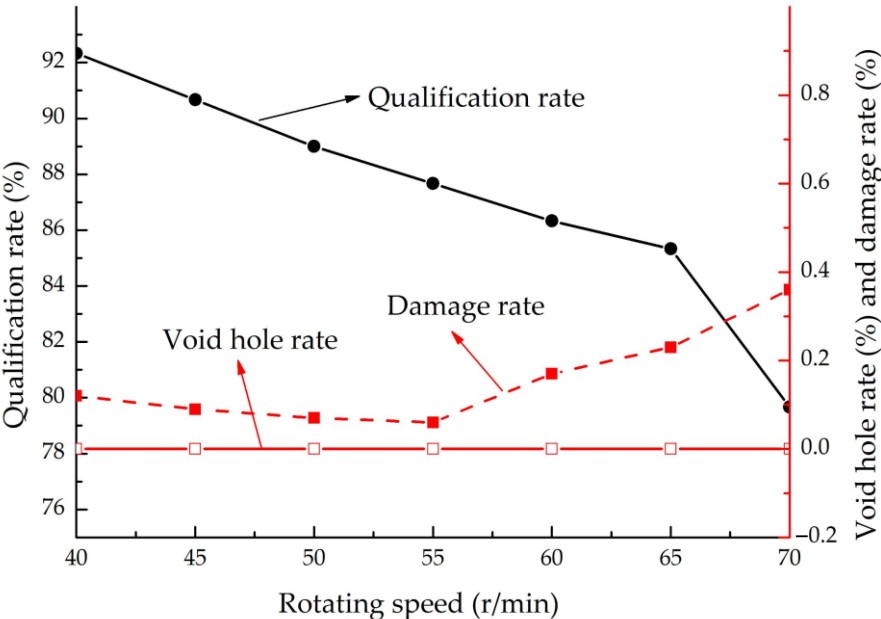

**Figure 15.** Trend chart of bench test results.

Based on the above experiment analysis, the single dibber wheel of the buckwheat hill-drop planter meets the DG/T007-2019 planter industry standard within the rotation speed range of 40–65 r/min, and the qualification rate is between 85.3% and 92.33%, the damage rate is less than 0.23%, and the void hole rate is 0. Through calculation, the driving speed range of the buckwheat hill-drop planter in this rotation speed range is 3.98–6.12 km/h. Therefore, in actual operation, the appropriate seeding speed can be selected according to actual needs.

## 4. Conclusions

1.  The structure of the hole forming device and the dimensions of its related components were determined. According to the hole distance of 170 ± 10 mm and the seed metering device radius of 210 mm, the duckbill upper jaw length was determined to be 65 mm, the duckbills number was 10, the pressure plate on the spring side length was 90 mm, the duckbill opening size was 8.79 mm, and the duckbill effective opening time was 0.1 s.
2.  Through EDEM and RecurDyn coupling simulation, the force range of the pressure plate was 95–102.6 N when the spring compression was 33.3 mm. When the rotation speed of the dibber wheel was 40 r/min, the contact time of a single duckbill with the soil was 0.2 s. Through calculation and theoretical analysis, the duckbill spring was selected, and the big end middle diameter of the spring was 36 mm, the small end middle diameter of the spring was 26 mm, the original length of the spring was 60 mm, and the wire diameter of the spring was 1.8 mm. Through bench tests of duckbill springs with wire diameters of 1.6 mm, 1.8 mm and 2.0 mm, the spring wire diameter of 1.8 mm was determined as the optimal test result.
3.  Through the single factor experiment of the dibber wheel rotation speed, it was concluded that when the designed dibber wheel rotated at 40–65 r/min, its seeding qualification rate was ≥85.3%, the void hole rate was 0, and the damage rate was less than 0.3%, which meted the industry standard DG/T007-2019 for seeders issued by the Ministry of Agriculture and Rural Affairs, PRC. The driving speed of the hill-drop planter is 3.98–6.12 km/h, which provides a speed reference for the field planting operations of the following planters.

**Author Contributions:** Writing—original draft, review, editing, and conceptualization, Y.C. (Yu Chen); software and investigation, Y.C. (Yuming Cheng); funding acquisition and project administration, J.C. (Jun Chen); resources and methodology, Z.Z.; data curation and formal analysis, C.H.; validation and supervision, J.C. (Jiayu Cao). All authors have read and agreed to the published version of the manuscript.

**Funding:** This work was funded by the National Key Research and Development Program of China (No.2018YFD0701100–2018YFD0701102), and the Key Research and Development Program of Shaanxi Province, China (No. 2019ZDLNY02-01).

**Conflicts of Interest:** The authors declare no conflict of interest.

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
