# Peer review of "Design and Experiment of the Buckwheat Hill-Drop Planter Hole Forming Device"

_agriculture, doi:10.3390/agriculture11111085_

Round 1
Reviewer 1 Report
The authors took up the topic related to agriculture, namely designed and verified a solution for forming holes in a buckwheat planter.
In their work, they presented a project with the necessary calculations, and screenshots from the simulation. All the results of the experiment are presented clearly in the form of tables.
In China, where there is the largest buckwheat cultivation area in the world, the subject matter is very important. However, it is not the only country where buckwheat is grown on a massive scale, so this solution should be universal and applicable to buckwheat planters used all over the world.
The conclusions from the research are correctly described.
My comments:
1. There is item 15 in the list of references, which is not in the text.
2. There is no space between the value and the unit in lines: 13-15, 18-19, 21, 23-24, 26, 188-189, 338, 370, 373, 404-405 and 413.
3. What do the numbers in brackets on lines 108-116 refer to?
4. Shouldn't there be figure 1a in line 108 instead of figure 1?
5. In lines 129, 137, 215, and 216, remove the space between the figure number and the subpoint (eg, figure 1 (b)), or if the Authors choose to leave a space, use it also in other references.
6. In summary, it would be worth referring to whether this solution can be used everywhere or only on a specific ground.
Author Response
Dear reviewer,
Thank you very much for your valuable and specified comments. We have made some changes accordingly. We hope these changes will meet with approval. The line number in this response is subject to that in the Word document. These changes are listed below.
Comments:
- There is item 15 in the list of references, which is not in the text.
Reply: The item 15 in the list of references has been added to the corresponding position of the manuscript.
- There is no space between the value and the unit in lines: 13-15, 18-19, 21, 23-24, 26, 188-189, 338, 370, 373, 404-405 and 413.
Reply: In the manuscript, the spaces between all numerical quantities and their units have been added. We have also revised the missing spaces in other places.
- What do the numbers in brackets on lines 108-116 refer to?
Reply: Thank you very much for your comment. We have revised Figure 1. The text in the figure has been replaced by numbers, and has a one-to-one correspondence with the names of the parts in the manuscript text (lines 108-116).
- Shouldn't there be figure 1a in line 108 instead of figure 1?
Reply: Yes, it’s should be figure 1(a) in line 108. We have revised the corresponding content in the manuscript.
- In lines 129, 137, 215, and 216, remove the space between the figure number and the subpoint (eg, figure 1 (b)), or if the Authors choose to leave a space, use it also in other references.
Reply: After careful comparison and modification, we have removed the space between the figure number and the subpoint. All the similar mistakes in the manuscript have been revised.
- In summary, it would be worth referring to whether this solution can be used everywhere or only on a specific ground.
Reply: Thank you very much for your comments. This study proposes a design and test method for the hole forming device of buckwheat hill-drop planter. Science the hole forming device with duckbill structure is widely used in hill-drop planters for wheat, cotton, peanuts, etc. The design, simulation and test methods proposed in this manuscript can also provide references for the hole forming device design of other crop planting machinery. Although the soil parameters were limited during the simulation, the method based on co-simulation analysis of discrete element software EDEM and multi-body dynamics software RecurDyn can be used in the hole forming device design of other crop planting machinery.
Sincerely,
CHEN, YU. (on behalf of all authors)
2021-10-26

Reviewer 2 Report
I would like to thank you for the work done on the article. It can be seen that the article has been well worked, both from the point of view of simulations and laboratory tests. However, the proof of the solution studied in real field tests is missing. Nevertheless, I congratulate you on your work and I would like to point out some observations that I would like to comment on:
- A number of limitations are raised in the simulations in which a series of assumptions are made, perhaps limited by the characteristics of the software used. However, I do not believe that these do not distort the study and, in addition, they have been corroborated with laboratory tests.
- Other suggestions are in line with the format:
o When indicating numerical quantities and their units in many occasions it does not perform spacing. For example, "36mm" instead of "36 mm" (line 21).
o On line 168 replace "amax" with "amax" and space the interval to improve readability.
o When referring to parts of a figure sometimes write them with spacing or without spacing, please unify criteria. For example, "Figure 3(b)" (line 171) and "Figure 6 (b)" (line 216).
o In figure 1, you list its different components or parts and indicate them with a number, however, this number does not appear in figure 1. In addition, you should improve the position of the text to improve legibility and clarity or, on the contrary, limit yourself to putting the numbers that you indicate in the text.
o In figure 6 a better positioning of the text and arrows should be made.
o In figure 7, eliminate the numerical values given by the 3D or increase the size so that there is no overlapping of these values in the image.
o In figure 12, the font size or scale should be increased so that it is easily readable.
o In the conclusions, on a personal note, I think it would be more appropriate to use periods or dashes to number them, since the numbers and letters in parentheses have been used for figures.
I believe that the article contributes to the advancement of knowledge in buckwheat planting and that it fulfills the goals initially stated.
Author Response
Dear reviewer,
We appreciate your valuable and insightful comments very much. We have tried our best to improve the manuscript and make further changes accordingly. We hope these changes will meet with approval. The line number in this response is subject to that in the Word document. These changes are listed below.
Comments:
- A number of limitations are raised in the simulations in which a series of assumptions are made, perhaps limited by the characteristics of the software used. However, I do not believe that these do not distort the study and, in addition, they have been corroborated with laboratory tests.
Reply: Thank you very much for your comments. Your opinion is very important to our research. In order to facilitate research and discussion, we made some assumptions and simplified the simulation scheme when carrying out some simulations. We will further study the limitations of these assumptions on the results of the study. The goal of this research is to propose a design and test method for the hole forming device of buckwheat hill-drop planter. Although some parameters and conditions were limited during the simulation, the method based on co-simulation analysis of discrete element software EDEM and multi-body dynamics software RecurDyn can be used in the hole forming device design of other crop planting machinery.Thank you again for your valuable comments to us.
- When indicating numerical quantities and their units in many occasions it does not perform spacing. For example, "36mm" instead of "36 mm" (line 21).
Reply: Thank you very much for your comments. There should be no spaces between the numerical quantities and their units, but referring to the template of this journal and the communication with the editor, we add all the spaces between the numerical quantities and their units in the manuscript.
- On line 168 replace "amax" with "amax" and space the interval to improve readability.
Reply: Thank you very much for the mistake you pointed out. “amax” has been replaced by “amax”, and the corresponding space has been added.
- When referring to parts of a figure sometimes write them with spacing or without spacing, please unify criteria. For example, "Figure 3(b)" (line 171) and "Figure 6 (b)" (line 216).
Reply: After careful comparison and modification, we changed all the similar mistakes in the manuscript to the form shown in Figure 3(b) when referring to parts of a figure.
- In figure 1, you list its different components or parts and indicate them with a number, however, this number does not appear in figure 1. In addition, you should improve the position of the text to improve legibility and clarity or, on the contrary, limit yourself to putting the numbers that you indicate in the text.
Reply: Thank you very much for the mistake you pointed out. We have revised Figure 1. The text in the figure has been replaced by numbers, and has a one-to-one correspondence with the names of the parts in the manuscript text.
- In figure 6 a better positioning of the text and arrows should be made.
Reply: We have revised Figure 6 to be better positioning of the text and arrows.
- In figure 7, eliminate the numerical values given by the 3D or increase the size so that there is no overlapping of these values in the image.
Reply: We have revised Figure 7 so that there is no overlapping of the values in the image.
- In figure 12, the font size or scale should be increased so that it is easily readable.
Reply: In figure 12, the font size has been increased.
- In the conclusions, on a personal note, I think it would be more appropriate to use periods or dashes to number them, since the numbers and letters in parentheses have been used for figures.
Reply: Thank you very much for your suggestion. We replaced the serial number in the conclusion with numbers and dashes.
Sincerely,
CHEN, YU. (on behalf of all authors)
2021-10-26
